# In Vitro Investigation of the Interaction of Avian Metapneumovirus and Newcastle Disease Virus with Turkey Respiratory and Reproductive Tissue

**DOI:** 10.3390/v15040907

**Published:** 2023-03-31

**Authors:** Frederik Bexter, Nancy Rüger, Hicham Sid, Alexandra Herbst, Gülsah Gabriel, Albert Osterhaus, Silke Rautenschlein

**Affiliations:** 1Clinic for Poultry, University of Veterinary Medicine Hannover, 30559 Hannover, Germany; 2Reproductive Biotechnology, TUM School of Life Sciences Weihenstephan, Technical University of Munich, 80333 Munich, Germany; 3Department of Viral Zoonoses—One Health, Leibniz Institute for Experimental Virology, 20251 Hamburg, Germany; 4Research Center for Emerging Infections and Zoonoses, University of Veterinary Medicine Hannover, 30559 Hannover, Germany

**Keywords:** oviduct organ culture (OOC), tracheal organ culture (TOC), avian metapneumovirus, newcastle disease virus, virus–host interactions, importin-alpha

## Abstract

In poultry, several respiratory viral infections lead to a drop in egg production associated with high economic losses. While the virus–host interactions at the respiratory epithelium are well studied, less is known about these interactions in the oviduct. To investigate possible differences between virus infections at these epithelial structures, we compared the interactions of two important poultry viruses on turkey organ cultures. Two members of the order Mononegavirales, the Avian Metapneumovirus (AMPV) and the Newcastle disease virus (NDV), were selected to conduct the in vitro experiments since these viruses can infect both the trachea and oviduct. In addition, we used different strains of these viruses, a subtype A and a subtype B strain for AMPV and the NDV strains Komarow and Herts’33, to detect possible differences not only between the tissues but also between different viral strains. Turkey tracheal and oviduct organ cultures (TOC and OOC) were prepared to investigate viral replication, antigen localisation, lesion development, and the expression pattern of interferon-λ and importin-α isoforms. All viruses replicated more efficiently in the oviduct than in the tracheal epithelium (*p* < 0.05). In addition, we observed higher expression levels of both, IFN-λ and importin-α in OOCs compared to TOCs. Our results indicated strain-dependent differences, with the AMPV-B- and Herts’33 strains being more virulent in organ cultures than the AMPV-A- and Komarow strains, based on the higher viral genome loads, more severe histological lesions, and higher upregulation of IFN-λ. Overall, our findings reveal tissue- and virus strain-dependent differences, which may have consequences for disease development in the host tissue and, subsequently, possible treatment strategies.

## 1. Introduction

Although various aspects of the virus–host interactions in the turkey respiratory tract have been investigated in vitro and in vivo by several groups [1,2,3,4], relatively little is known about the virus–host interactions at the epithelium of the reproductive tract.

Therefore, the main objective of this study was to investigate the virus–host interaction at the epithelial surfaces, comparing the effects of virus infections on the turkey trachea and oviduct. Organ cultures were chosen as infection models since both tracheal and oviduct organ cultures have successfully been established to investigate viral infections [5,6,7,8]. This ex vivo technique not only reduces the need for animal experiments but also allows experiments to be conducted under standardised conditions with comparable sample sizes, which is difficult to achieve in an in vivo setting. Moreover, due to an intact cell network, organ cultures enable the investigation of various effector mechanisms of the innate immune system [9,10].

Within the order of Mononegavirales, two economically relevant viruses of poultry, the Avian Metapneumovirus (AMPV) and the Newcastle disease virus (NDV), were chosen for the infection experiments [11,12] since both may induce lesions in the respiratory as well as in the reproductive tract of turkeys [13,14,15]. The selected viruses have a non-segmented genome with single-stranded, negative-sense RNA [16].

AMPV belongs to the family *Pneumoviridae*, genus *Metapneumovirus*, and species *Avian metapneumovirus* [16]. So far, four different subtypes (A–sD) have been detected in poultry, of which, subtypes A and B are the most common ones in Europe [17]. Subtype C is mainly found in North America and is the subtype most closely related to the Human Metapneumovirus (HMPV), based on phylogenetic analysis [17]. Recently, two new subtypes were suggested, which were isolated from a parakeet and a gull [18,19]. AMPV is mainly known to cause respiratory diseases, such as turkey rhinotracheitis (TRT) in turkeys and swollen head syndrome (SHS) in chickens [13]. AMPV-infected birds show typical respiratory symptoms, such as dyspnea, oculonasal discharge, coughing, and swollen infraorbital sinuses [4]. Based on flock morbidity rates and viral replication in tracheal tissues, turkeys are more susceptible to AMPV infections than chickens [2,13].

NDV belongs to the family *Paramyxoviridae*, genus *Orthoavulavirus*, and species *Avian orthoavulavirus-1* [20]. Based on the mean death time of chicken embryos and the intracerebral pathogenicity index (ICPI), NDV strains can be divided into four groups: apathogenic, lentogenic, mesogenic, and velogenic [21]. Infections with lentogenic NDV are normally subclinical, whereas, in young birds, mild respiratory symptoms may be observed [22]. Mesogenic strains usually cause respiratory disease with low mortality rates. Severe, acute infections with high mortality rates (up to 100%) are caused by velogenic NDV and are often associated with haemorrhages in the gastrointestinal tract [22]. Further, neurotropic strains have been described, and infected birds may show typical neurological signs, such as tremors, torticollis, and paralysis [23]. Furthermore, both AMPV and NDV may affect the reproductive tract, which is associated with a drop in egg production and disorders in eggshell quality in laying birds [11,24].

To differentiate not only between these two viruses but also to identify possible virus strain variations, we used two different strains of each AMPV and NDV. Therefore, we selected AMPV subtype A and subtype B strains as well as the mesogenic (Komarow) and the velogenic (Herts’33) strains of NDV. To study the virus–host interaction of either AMPV or NDV with the respiratory or reproductive epithelium, the following parameters were analysed: virus replication, the distribution of the viral antigen, and histopathological lesion development. The similarity between histological structures of the respiratory and reproductive epithelium with ciliated cells enables a high degree of comparability between these tissues [2,6].

To investigate possible tissue-dependent variations in the host immune response following infection with either the AMPV or NDV strains, we examined the mRNA expression pattern of interferon-lambda (IFN-λ) and importin-α isoforms. Due to its specific receptor distribution at the epithelial surface, IFN-λ has been shown to be a key antiviral factor at the mucosal surfaces [25,26,27]. Recent studies have demonstrated the increased expression of IFN-λ after infection with different respiratory viruses in vitro and in vivo [28].

Importin-α is located at the nuclear cell membrane and is responsible for the intranuclear cell transport of various proteins [29]. This importin-based pathway is used by different viruses to enter the nucleus for viral replication [30,31]. Although NDV and AMPV replicate in the cytoplasm, we assumed interactions between these viruses and importins. For NDV, it has been described that importin-α5 plays an inhibitory role in the nuclear transport of the NDV matrix protein [32]. Since several studies described the presence of the M protein in the nucleus and its positive regulatory effects on viral replication, we speculate an indirect regulatory effect of importin-α on NDV replication [33,34,35,36].

The results of this study enhance the understanding of interactions between AMPV and NDV with different mucosal structures and might help to improve intervention strategies against these viruses.

## 2. Materials and Methods

### 2.1. Viruses

The AMPV subtype A strain BUT 8544 (AMPV-A) and the AMPV subtype B Italy strain (AMPV-B) were kindly provided by Dr. R.C. Jones in Liverpool, UK. The mesogenic Komarow NDV strain (NDV-KO) and the velogenic Herts’33 NDV strain (NDV-HS) were derived from the strain collection of the Clinic for Poultry at the University of Veterinary Medicine in Hannover, Germany.

The AMPV strains were propagated in TOCs, and both of the NDV strains were propagated in specific-pathogen-free (SPF) chicken embryos following standard procedures [4,37]. Titers were calculated according to Reed and Muench [38].

### 2.2. Experimental Design

A total of eight in vitro experiments were conducted, including one experiment per virus strain per tissue plus an independent repeat. Tracheal and oviduct organ cultures were inoculated separately with one of the four different virus strains (AMPV-A, AMBV-B, NDV-HS, or NDV-KO).

The rings of both tissues were randomly divided into infection and virus-free control groups (n = 3 to 5 rings/treatment/time point). Each organ culture ring was inoculated with 100 µL of the 10^3^ 50% ciliostatic doses (CD50) of each virus/mL. The virus-free group was treated with 100 µL of the virus-free medium. After one hour of incubation, 900 µL of the medium was added without removing the inoculum.

Samples were taken at five consecutive time points (1, 24, 48, 72, and 96 h post-inoculation = hpi) for each experiment to investigate viral replication, the distribution of the viral antigen, the development of histopathological lesions, and the expression pattern of IFN-λ as wells as importin-α 1, 3, 4, 5, and 7.

As a vital parameter of the organ culture rings, the ciliary activity was checked daily using microscopy.

The results represent the summarised data from one representative experiment. All birds in this study were sacrificed humanely according to the animal welfare regulations of the German animal welfare law (permission number: TiHo-T-2019-12).

### 2.3. Ciliostasis Assay

Ten tracheal organ culture rings with 100% ciliary activity were selected and randomly distributed among the infection groups. For ten days post-inoculation (dpi), the ciliary activity of each ring was monitored daily using an inverted microscope (Zeiss, Jena, Germany). Subsequently, the average ciliary activity of the ten rings, per day, and the infection group was calculated [39]. The ciliostasis assay was only performed in TOCs since the structure and architecture of the oviduct do not allow for the assessment of total ciliary activity.

### 2.4. Tracheal Organ Cultures (TOCs)

TOCs were prepared one day before the hatching of 27-day-old turkey embryos of a commercial genotype (B.U.T 6, Moorgut Kartzfehn, Bösel, Germany). After removing the connective and muscle tissue from the trachea, the organ was cut into rings aseptically. Each tracheal ring was transferred into a 5 mL tube with 1 mL of the prewarmed medium before incubation in an overhead rotator at 37 °C (Heidolph Instruments, Schwabach, Germany) [39,40,41]. Four days after preparation, only the rings with 100% ciliary activity were selected for the in vitro experiments. Five rings per time point and treatment condition were randomly assigned to the different groups.

### 2.5. Oviduct Organ Cultures (OOCs)

The OOCs were prepared from the juvenile oviduct of 19-week-old female turkeys (B.U.T 6).

Since it was difficult to differentiate between the five functional segments of the juvenile oviduct, rings were taken from the middle third of the organ. Based on preliminary experiments, we found that the OOCs had a shorter viability period compared to TOCs. Therefore, the inoculations of the OOCs were performed directly on the day of preparation. The OOCs were incubated in 24-well cell culture plates on a vibrating laboratory shaking plate (GFL, Burgwedel, Germany) at 37 °C with 5% CO_2_ [5,6]. At least three to five rings were collected for each investigated time point and group.

### 2.6. Histopathological Lesion Development

For the detection of lesion development, three to five rings were collected per infection group and time point for each conducted experiment. Organ culture rings were fixed in 4% paraformaldehyde (PFA) and embedded in paraffin. Then, 2 µm sections were prepared and stained with hematoxylin and eosin (H&E) for microscopical examination [42]. The following histopathological lesions were investigated: ciliary loss, the flattening and detachment of epithelial cells, and submucosal oedema [2,39].

### 2.7. Antigen Detection Using Immunohistochemistry

Paraffin-embedded tissue sections were first treated with xylol, followed by descending alcohol dilutions, according to standard procedures [43]. Subsequently, slides were incubated in sodium citrate buffer at 50 °C for 16 h, followed by a permeabilisation step with 0.2% Triton™ X-100 (Merck, Darmstadt, Germany). Bovine serum albumin (BSA) (Roth, Karlsruhe, Germany) was used at a concentration of 3% in phosphate-buffered saline (PBS) for blocking.

For AMPV antigen staining, a polyclonal chicken anti-AMPV A antiserum, collected from AMPV hyperimmunised chickens, was used at a 1:100 dilution [2]. For NDV antigen staining, a commercial polyclonal chicken anti-NDV antibody (Biozol, Eching, Germany) was utilised at a 1:100 dilution. The sections were incubated with the primary antibody in a humid chamber at 4 °C for 16 h, followed by a washing step with PBS. Subsequently, a goat anti-chicken IgG-HRP-labelled second antibody (Biozol) was added at a dilution of 1:1000, and slides were incubated at 37 °C for 75 min [39].

Finally, staining was conducted with a 3.3′-diaminobenzidin-substrate (DAB; Vector Laboratories, Inc., Burlingame, CA, USA). Counterstaining was performed with haemalaun, and slides were mounted with Aquatex^®^ (Merck, Darmstadt, Germany). All stained sections were examined microscopically (Leica Microsystems, Wetzlar, Germany) with 400-fold magnification.

### 2.8. Quantitative Real-Time PCR (qRT-PCR) for Virus Detection, Interferon-λ and Importin-α Isoform Quantification

Each organ culture ring was homogenised with the Precellys^®^ 24-tissue homogeniser (Bertin Technologies, Montigny-le-Bretonneux, France). Subsequently, RNA isolation was accomplished using the innuPREP^®^ RNA mini kit (Analytik Jena GmbH, Jena, Germany).

The quantification of the viral genome and IFN-λ mRNA expression was determined using the qScriptTM XLT 1-Step RT-qPCR ToughMix Low ROX-polymerase (Quantabio, Beverly, MA, USA) [44,45]. The mRNA expression of different importin-α isoforms was investigated using the Luna Universal Probe One-Step PCR-Kit (New England BioLabs GmbH, Frankfurt am Main, Germany) following the manufacturer’s instructions. Each sample was tested in duplicates and normalised against the 60S ribosomal protein L13 (RPL13) housekeeping gene of the respective sample, as described previously [46,47]. The viral load is presented as the mean of the cycle threshold (CT) values. The expression of IFN-λ and importin-α isoforms is shown as the log 2-fold change in relation to the normalised data (∆CT) of the virus-free control group. All primers and probes used in this study are listed in Appendix A.

### 2.9. Statistical Analysis

The Shapiro–Wilk normality test was used to test the data for normal distributions. Statistically significant differences between the time points per group were verified using the Tukey HSD all-pairwise comparisons test (ANOVA, with α = 0.05) and verified using two-sample t-tests comparing two consecutive time points. The significant differences between the virus-infected groups in comparison to the virus-free control group were also tested using the two-sample t-test. All tests were conducted with Statistix, Version 10.0 (Analytical Software, Tallahassee, FL, USA). For multiple comparisons of the not normally distributed data between the inoculated groups and virus-free controls, the Kruskal–Wallis all-pairwise comparison test was used (α = 0.05). Differences were considered significant at a *p*-value < 0.05. All graphs were created with GraphPad Software Prism 9 (version 9.2.0, San Diego, CA, USA).

## 3. Results

### 3.1. Viral Genome and Antigen Detection in TOCs and OOCs

#### 3.1.1. Virus Quantification Using qRT-PCR

The viral genome was quantified using qRT-PCR to identify possible differences not only between tissues but also among both viruses and different viral strains. We focused on the early phase of infection and investigated five consecutive time points (1–96 hpi) (Figure 1 and Appendix A).

The viral loads of both AMPV strains and of the NDV-KO strain increased significantly between one and 24 hpi (*p* < 0.05) (Appendix A). In both tissues, the viral genome loads reached their peak at 48 hpi, with a subsequent plateau phase in both tissues, except from NDV-HS in OOCs (Figure 1).

The comparisons of both AMPV strains revealed that the replication pattern varied slightly and that subtype B loads were, overall, significantly higher in the TOCs and OOCs compared to AMPV subtype A (*p* < 0.05) (Figure 1A,B). Differences in the replication patterns between the NDV strains were predominantly detected in the oviduct (Appendix A). While both NDV strains reached their peak at 48 hpi in the TOCs, the replication pattern differed in the OOCs, where the viral load in the Herts‘33 group continued to increase up to 96 hpi (Figure 1C,D).

Overall, comparable viral replication patterns were detected in TOCs and OOCs. However, compared to the OOCs, the detected viral genome quantities were significantly higher in the TOCs, particularly for AMPV-B and NDV-KO (*p* < 0.05) (Figure 1B,C).

Despite comparable trends in the replication patterns of TOCs and OOCs between repeat experiments, variations in the total viral loads were observed for all four viruses, as exemplified in Figure 2 for AMPV-B.

#### 3.1.2. Immunohistochemical Detection of the Viral Antigen

Immunohistochemical staining was performed to evaluate the localisation of the viral antigen in the TOC and OOC. AMPV-A- and NDV-HS-infected organ culture rings were selected as representative results for all virus strains (Figure 3).

The viral antigen of both AMPV and NDV strains was detected at the cilia and the epithelial cell border in both tissues. Declines in antigen-positive epithelial cells and cilia were observed at later time points, accompanied by ciliary loss and epithelial cell flattening. 

### 3.2. The Effect of Viral Infections on the Functional and Histological Structures of Epithelial Cells

#### 3.2.1. Ciliostasis Assay

To determine possible differences in virus-induced ciliostasis, induced by either AMPV or NDV, ten TOC rings per group were microscopically evaluated for ten days post-inoculation (dpi). Overall, 100% ciliary activity was detected in TOCs from all groups on the first dpi (Figure 4). For all infected groups, the onset of ciliostasis started at 2 dpi. Thereafter, ciliary activity in AMPV-B-infected TOCs was significantly more reduced compared to subtype A at all consecutive time points (*p* < 0.05). At 3 to 5 dpi, a significantly faster progression of ciliostasis was observed in the NDV-KO group in comparison to NDV-HS (*p* < 0.05). Although differences in the progress of ciliostasis were revealed between the viral strains, both NDV strains and AMPV-B followed a similar trend. For these three viruses, complete ciliostasis was observed at 8 dpi. In contrast, the ciliary activity in the AMPV-A group decreased moderately and remained at almost 60% up to 10 dpi (Figure 4).

Since observations were similar between repeat experiments, data from one representative experiment are presented in Figure 4.

#### 3.2.2. Histopathological Lesion Development

Histopathological lesions were investigated in virus-infected TOCs and OOCs. No lesions were observed in virus-negative tissues.

Ciliary loss was detected in both tissues after infection with either AMPV or NDV starting at 24 hpi in TOCs and at 48 hpi in OOCs (Figure 5). In AMPV-A-infected TOCs, both ciliary loss and epithelial flattening were observed at later time points and in fewer tissue samples compared to the other viruses. Additionally, no cell detachment was observed in AMPV-A-infected TOCs (Appendix A). Further, neither epithelial flattening nor oedema occurred in the oviduct after AMPV-A infection. The development of oedema at 48 to 96 hpi was similar in the three other virus-infected TOCs. While both NDV strains caused the detachment of epithelial cells as early as 48 hpi; this lesion was first detected at 72 hpi in AMPV-B-infected TOCs (Appendix A). The main difference concerning epithelial cell flattening was observed between NDV strains in OOCs. All Herts‘33-infected rings were affected at 72 hpi, while only two of five OOCs showed this lesion in the Komarow group (Appendix A).

Overall, the ciliary loss was comparable in virus-infected TOCs and OOCs. Furthermore, the onset of epithelial flattening and oedema occurred later in the oviduct than in TOCs, starting at 72 hpi and 96 hpi, respectively. In comparison to tracheal tissue, no cell detachment was found in OOCs. As representative results for all viruses, the histopathological lesions of the AMPV-A- and NDV-HS-inoculated TOCs and OOCs are presented in Figure 5.

### 3.3. The Effect of Different Viral Infections on the Host Response

#### Quantification of IFN-λ mRNA Expression

The IFN-λ mRNA expression levels were detected in TOCs (Figure 6) and OOCs using qRT-PCR (Figure 7).

A significant upregulation of IFN-λ was shown for AMPV-B in both tissues compared to subtype A (*p* < 0.05). In TOCs, the infection with Herts‘33 caused a subsequent upregulation starting at 48 hpi (Figure 6C). Contrarily, in Komarow-infected TOCs, a constant downregulation of IFN-λ mRNA expression was observed (Figure 6D). In the oviduct, both NDV strains were associated with a continuous increase in IFN-λ expression levels compared to virus-free OOCs, peaking at 72 hpi and decreasing thereafter (*p* < 0.05) (Figure 7C,D).

Comparing the tissues, higher levels and more upregulation of IFN-λ were observed in the oviduct than in the trachea. In particular, infections with AMPV-B and NDV led to significant increases in IFN-λ levels in the OOCs at 72 hpi (*p* < 0.05). In both tissues, the upregulation of IFN-λ coincided with the viral load plateau phase (Figure 6 and Figure 7).

## 4. Quantification of Importin-α Isoform mRNA Expression

The mRNA expression of five different importin-α isoforms (IMP-α1, IMP-α3, IMP-α4, IMP-α5, and IMP-α7) was compared in TOCs and OOCs after AMPV or NDV inoculation. All five importin-α isoforms were expressed in both tissues. In both, IMP-α1 and IMP-α3 had the lowest and the highest expression levels, respectively (Appendix A). Compared to the TOCs, expression levels were higher in the OOCs for all importin-α isoforms, especially for IMP-α1 and IMP-α7 (*p* < 0.05).

In virus-infected TOCs, the downregulation of all investigated importin-α isoforms was dominant, being most evident for IMP-α1 and IMP-α7 (Figure 8). The importin-α expression levels in either AMPV-A- or AMPV-B- infected TOCs (Figure 8A,C) showed less downregulation compared to NDV-infected TOCs (Figure 8E,G). The peaks of downregulation were either at 72 or 96 hpi. The significantly lowest interferon-α levels were detected in NDV-HS-infected TOCs compared to the other viruses (*p* < 0.05) (Figure 8E). Significantly decreased IMP-α1 and IMP-α7 levels, related to the virus-free group, were also encountered in the NDV-HS group (*p* < 0.05).

In the oviduct, a significant upregulation of most importins was noted after inoculation with AMPV-A as well as with both NDV strains (*p* < 0.05). Overall, the highest upregulation in comparison to the virus-free group was found in the Herts’33 group, with a peak at 72 hpi (Figure 8F). Significantly enhanced importin-α mRNA expression levels were also observed in AMPV-A- compared to subtype B-infected OOCs at 72 hpi (*p* < 0.05) (Figure 8B,D). In NDV-KO-infected OOCs, the highest importin-α expression levels were detected at 24 hpi compared to the virus-free control (Figure 8H). In OOCs, both NDV strains were associated with an initial tendency of upregulation followed by a significant decline of importin-α expression at 96 hpi (*p* < 0.05).

## 5. Discussion

The present study aimed to expand the knowledge of pathogen–host interactions at ciliated epithelia by comparing the respiratory and reproductive tracts of turkeys. While this has been investigated extensively in the trachea, less is known about the oviduct. Since it is difficult to compare these epithelial surfaces simultaneously in vivo, we used organ cultures as an in vitro infection model. Both tissues were inoculated with one of two subtype strains of AMPV or one of two NDV strains to detect possible differences not only between virus species but also to see virus strain variations. Therefore, we quantified the viral genome and assessed the antigen localization in the tissues. Further, we examined histopathological lesions with a focus on ciliary activity. We investigated the mRNA expression pattern of IFN-λ patterns as an important epithelial-associated innate immune response to viral infections, as well as the expression levels of five different importin-α isoforms associated with virus transport between cell compartments or innate immunity [25,29].

The qRT-PCR results highlight the successful viral replication of all investigated AMPV and NDV strains in both organ cultures. Active AMPV replication in the oviduct has only been demonstrated twice in previous in vivo studies [14,48]. To our knowledge, it is the first time that successful viral replication of AMPV-A has been demonstrated in the turkey oviduct. Therefore, our results clearly demonstrate that AMPV is not only able to infect respiratory but also reproductive tract epithelial cells and may directly contribute to disorders at these sites. These findings suggest that the appearance of reduced egg production and thin-shelled eggs can be directly related to AMPV infection, rather than to a secondary effect. Although viral replication patterns with an increase up to 48 hpi and a subsequent plateau phase were similar, we detected differences in viral genome quantities between tested strains. In the case of AMPV, higher viral loads were observed in AMPV-B-infected organ cultures compared to subtype A (*p* < 0.05). Similarly, NDV-HS quantities were higher in comparison to NDV-KO (*p* < 0.05), which is possibly associated with virulence differences. Overall, in both tissues, AMPV-A had the lowest mRNA expression levels of all the investigated viruses. Several in vitro studies investigated either subtype A or B [2,39] but a comparison of different subtypes has only been performed in vivo twice [1,49]. Our results showed that the investigated AMPV-B strain can replicate more efficiently in TOCs as well as in OOCs compared to the AMPV-A strain and, therefore, we suggest that this strain may have a higher virulence than AMPV-A under in vitro circumstances. In contrast, previous in vivo experiments determined higher viral loads and severe clinical signs in AMPV-A-infected turkeys compared to AMPV-B [1,49]. Together, these findings suggest the need to conduct further experiments in birds as well as in vitro to investigate possible AMPV strain variations.

Interestingly, we found significantly higher viral loads in TOCs compared to OOCs for each virus strain (*p* < 0.05). Therefore, we assume a higher susceptibility of the respiratory tract, in contrast to the reproductive tract, to these viral infections. By using organ culture infection models, the higher susceptibility of the trachea compared to the oviduct has also been demonstrated for different infectious bronchitis virus strains in the past [50]. Since the adaptive immune response has no impact on our infection model, we assume tissue-dependent differences in the innate immune response. Preceding studies have described sialic acid receptors as attachment sites for NDV and HMPV to initiate the infection [51,52,53]. Since AMPV is closely related to HMPV, we presume similar infection mechanisms [54]. Therefore, we speculate that different distributions of lectin-binding sites in the trachea compared to the oviduct could explain the different virus quantities.

In the present study, antigen detection through immunohistochemistry was confirmed by the PCR data, depicting the attachment of each investigated virus strain. We found the viral antigens, either of AMPV or NDV, located at the cilia and the apical cell membrane in the trachea as well as in the oviduct, which has previously been shown for AMPV [2,39,48] in poultry and for NDV in the oviduct [24]. The results of this study confirmed cilia as the main attachment structure for AMPV and NDV infections.

A reduction in ciliary activity is a common consequence of pathogen–host interactions at the epithelial surfaces associated with several respiratory viruses [55,56,57,58]. Ciliary movement is part of the mucociliary clearance in the trachea, a key defence mechanism against invading respiratory pathogens and inhaled molecules [59]. Due to dysfunctional clearance, the attachment of infectious particles is enhanced, which contributes to severe tissue damage as well as the exacerbation of the diseases [60]. The ciliostasis assay results of the current study revealed that AMPV-A had a significantly lower effect on ciliostasis compared to AMPV-B (*p* < 0.05). A similar decline in ciliary activity was observed for AMPV-B and both NDV strains, with a reduction in ciliary movement by 50% at 4 dpi and 100% at 7 dpi. Comparable effects of AMPV on ciliary activity have been shown in chicken TOCs. Although NDV-KO is classified as less virulent than NDV-HS, this virus-induced significantly enhanced ciliostasis between 2 dpi and 5 dpi compared to NDV-HS, supporting the differences seen in pathogenesis in vivo [61].

Since both of the investigated tissues have ciliated epithelial cells, we hypothesised to find similar histopathological lesions [4,62]. Indeed, ciliary loss, the flattening of epithelial cells, and submucosal oedema were observed in both tissues, whereas cell detachment was only found in TOCs. Further, we observed differences concerning the extent and time points of pathological changes between the tissues and viruses.

Ciliary loss was confirmed in both AMPV- and NDV-infected TOCs and OOCs through histopathological examination, as shown before [24,39]. AMPV-A infection was associated with fewer histopathological lesions compared to AMPV-B and both NDV strains, correlating with the ciliostasis assay results. In TOCs, more NDV-HS-infected rings developed cell detachment compared to NDV-KO. In OOCs, the time point of detectable flattened epithelial cells was earlier in the NDV-HS group compared to the other NDV strain. These findings support the hypothesis that more virulent NDV strains cause more severe epithelial damage. When comparing both tissues, the loss of cilia was similar, whereas all other investigated lesions occurred at delayed rates and less frequently in OOCs.

Therefore, the more severe histopathological lesions in the trachea coincide with the qRT-PCR results, suggesting that tracheal tissue is more susceptible to either AMPV or NDV infection than the oviduct epithelium.

Several studies have revealed the importance of IFN-λ in protecting the host against viral infections, especially at epithelial barriers [26,63,64]. Sid et al. observed increased IFN-λ levels after avian influenza infection in both TOCs and OOCs [65]. In our study, we observed higher IFN-λ mRNA expression levels in TOCs and OOCs infected with either AMPV-B or NDV-HS compared to subtype A and NDV-KO, respectively. Together with the detection of higher viral loads and severe histological lesions, these findings indicate an enhanced virulence of AMPV-B and NDV-HS. Liman et al. observed a higher interferon expression after AMPV-B infection compared to subtype A in vivo [1]. Comparing TOCs and OOCs, significantly higher IFN-λ expression levels were detected in the oviduct, with particularly higher cytokine loads at time point 72 hpi compared to the virus-free group (*p* < 0.05). The increased IFN-λ mRNA expression correlated with the plateau phase of the viral replication curves, which could possibly be explained by its antiviral effect [25,26]. The enhanced IFN-λ mRNA expression and less viral replication in OOCs compared to TOCs support this assumption. More studies are needed to better understand the antiviral mechanisms of IFN-λ at epithelial barriers in turkeys,

Additionally, we investigated the effect of AMPV and NDV infection on importin-α regulation in TOCs and OOCs. To our knowledge, this study is the first to investigate the expression pattern of different importin-α isoforms in the turkey oviduct. The critical role of importin-α3 and -α7 for the viral entrance in the nucleus, and its subsequent replication, has previously been shown for the avian influenza virus (AIV) [30]. In contrast, a negative regulatory effect on the intranuclear NDV import has been described for importin-α5 [32]. Unique to the current study is the investigation of the possible effect of AMPV infections on the expression pattern of different importin-α isoforms.

Our results clearly show that importin-α isoforms are differentially expressed in both tissues after infection with the various viruses. In TOCs, we mainly detected the downregulation of importins, in contrast to upregulation in OOCs. Comparing the two virus species, NDV-infected organ cultures showed a significantly higher degree of up- and downregulation of importin-α isoforms compared to AMPV (*p* < 0.05). Due to the display of data as log 2-fold changes, we have to take into account that values higher or lower than two probably have no biological relevance. Therefore, we determined only a minor effect of AMPV infections on importin-α regulation in the organ culture model. In NDV-infected TOCs, importin-α1 and α were significantly more downregulated compared to the other importins (*p* < 0.05). In OOCs infected with either of the two AMPV strains or NDV-HS, each importin-upregulation peaked at 72 hpi. In contrast, the differential importin-α mRNA expression levels in NDV KO-infected OOCs peaked at 24 hpi. Overall, our results reveal tissue- and virus strain-dependent differences in importin-α regulation. Next to its transport function, several studies described other functions of importin-α, such as the promotion of intranuclear viral replication, nuclear membrane assembly, proteasomal function, and cell mitosis [66,67]. Therefore, further studies are needed to gain new insights into the biomolecular mechanisms of importins in the context of AMPV and NDV infections by focusing on their involvement in viral transport, viral replication, and host defence. In addition, it would also be useful to investigate importin-β since the interaction of importin-α and -β is essential for the intranuclear transport mechanism [29,68,69]. In particular, for NDV, it is known that importin-β1 is involved in transporting the matrix protein into the nucleus [32].

## 6. Conclusions

Our study clearly demonstrates that in vitro experiments in organ cultures represent a valuable model to investigate pathogen–host interactions at epithelial surfaces. Taken together, the higher viral load and more severe histopathological lesion development suggest a higher susceptibility of the respiratory epithelium to AMPV and NDV infections compared to the oviduct. Furthermore, our results point out strain-specific differences, with AMPV-B and NDV-HS being more virulent than the other investigated viruses. This study further reveals significant differences in the innate immune responses between the trachea and the oviduct. Expression levels of IFN λ and importin-α are more enhanced in OOCs compared to TOCs.

Our study is the first to investigate the mRNA expression patterns of different importins in the turkey oviduct and the influence of AMPV infection on importin-α regulation. These findings suggest possible new approaches for intervention strategies at epithelial surfaces in poultry. Finally, our results confirm differences between tissues in both the viral interaction with the epithelia and the host response. Therefore, further detailed investigations are needed to understand more about the consequences of these different pathogen–host interactions at epithelial structures.

## Figures and Tables

**Figure 1 viruses-15-00907-f001:**
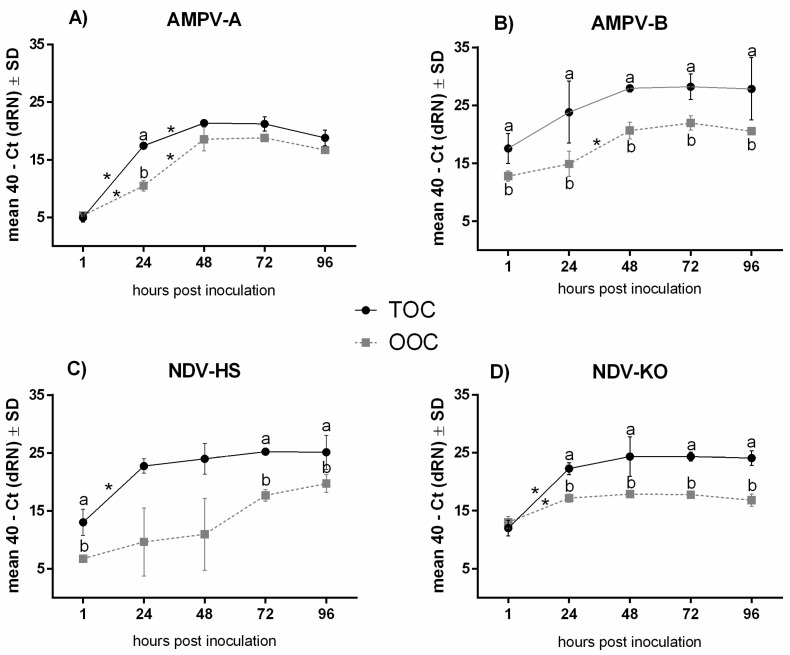
Comparative quantification of viral genome using qRT-PCR in TOCs and OOCs after inoculation with AMPV-A (**A**), AMPV-B (**B**), NDV-HS (**C**), or NDV-KO (**D**). For each group, three to five rings were collected at 1, 24, 48, 72, and 96 hpi and processed for viral quantification. Normalised data are presented as mean 40 - Ct. Error bars represent standard deviations (SD). * indicates significant differences between different inoculation time points for each virus, determined using the Tukey HSD all-pairwise comparison test. Small letters indicate significant differences between TOCs and OOCs at the same time point for each virus, determined using two-sample *t*-tests. *p*-value < 0.05. Graphs represent data from one representative experiment.

**Figure 2 viruses-15-00907-f002:**
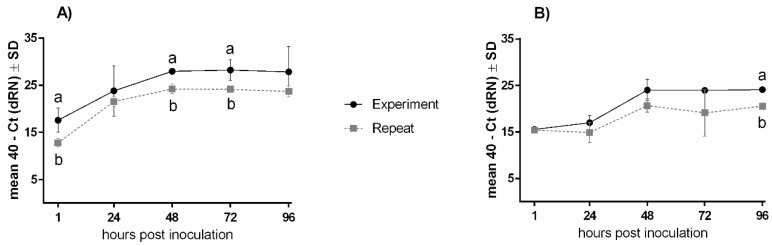
Comparison of AMPV-B viral genome in TOCs and OOCs between two experiment series. For each group, three to five rings were collected at 1, 24, 48, 72, and 96 hpi and processed for AMPV-B quantification in TOCs (**A**) and OOCs (**B**) using qRT-PCR. Normalised data are presented as mean 40 - Ct. Error bars represent standard deviations (SD). Small letters indicate significant differences between the two experiments for either the trachea or oviduct at the same time point (**A**,**B**), determined using two-sample *t*-tests. *p*-value < 0.05.

**Figure 3 viruses-15-00907-f003:**
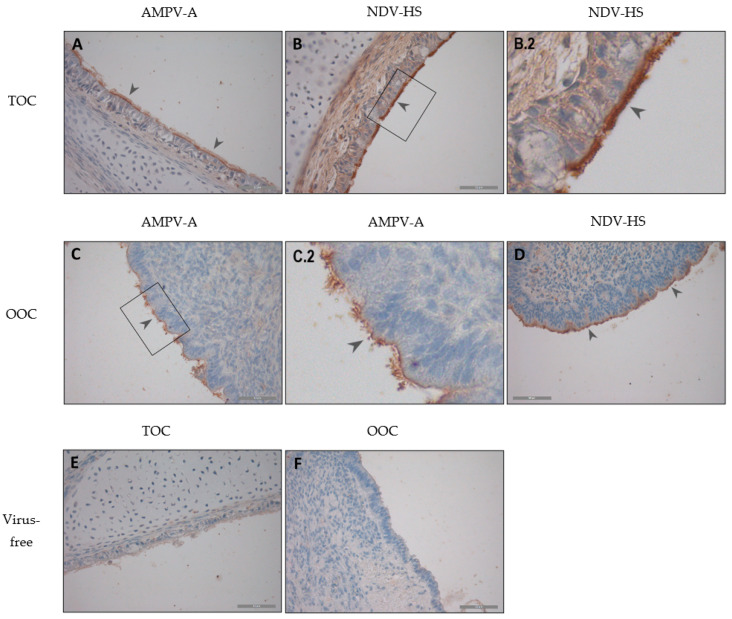
Antigen detection of AMPV-A (**A**,**C**) and NDV-HS (**B**,**D**) in TOCs (**A**,**B**) and OOCs (**C**,**D**) through immunohistochemistry. Brown staining indicates virus-infected cells. (**A**) TOC infected with AMPV-A, (**B**) TOC infected with NDV-HS, (**C**) OOC infected with AMPV-A, (**D**) OOC infected with NDV-HS, (**E**) virus-free TOC, (**F**) and virus-free OOC—all at 48 hpi. Arrowheads mark specific viral antigen staining, and black boxes visualize zoomed-in areas (shown in **B.2**,**C.2**). The scale bar was 50 µm.

**Figure 4 viruses-15-00907-f004:**
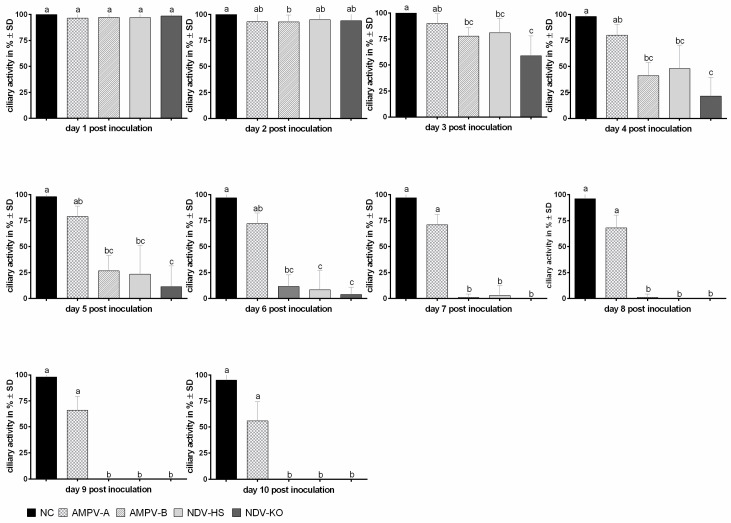
Ciliostasis in TOCs after inoculation with either AMPV-A, AMPV-B, NDV-KO, or NDV-HS. The daily average ciliary activity for ten TOCs per group is displayed. Error bars indicate standard deviations (SD). Small letters represent significant differences between the four different virus strains and the negative control at the same time points, at *p* < 0.05, using the Kruskal–Wallis all-pairwise comparison test. Graphs represent data from one representative experiment.

**Figure 5 viruses-15-00907-f005:**
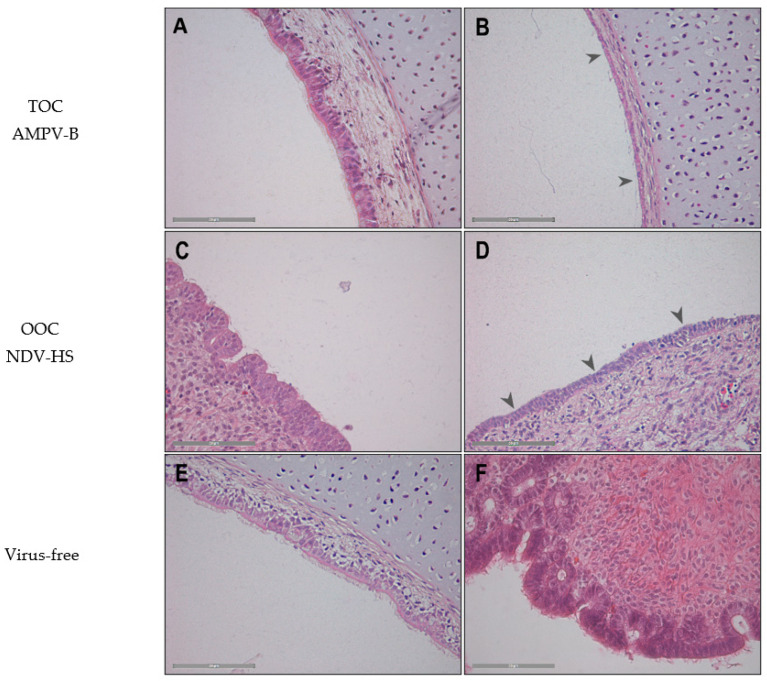
Histopathological lesion development at the epithelial surface of TOCs (**A**,**B**) and OOCs (**C**,**D**). Images are AMPV-B-infected TOCs at (**A**) 24 hpi and (**B**) 72 hpi, and NDV-HS-infected OOCs at (**C**) 24 hpi and (**D**) 96 hpi. (**E**) TOC and (**F**) OOC illustrate virus-free rings at time point 96 hpi. Arrowheads highlight cilia loss. The scale bar is 50 µm.

**Figure 6 viruses-15-00907-f006:**
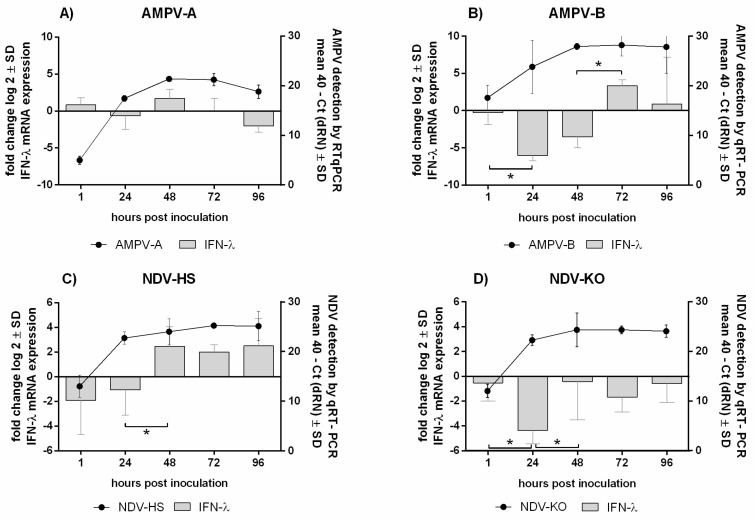
Quantification of IFN-λ mRNA expression in correlation with AMPV and NDV replication in TOCs. (**A**) AMPV-A, (**B**) AMPV-B, (**C**) NDV-HS, and (**D**) NDV-KO. Five rings/group/time points were investigated. Error bars indicate standard deviations. * indicate significant differences between the virus-inoculated group and the virus-free group at the same time point, determined using the Tukey HSD all-pairwise comparison test (ANOVA, *p* < 0.05). The graphs show data from a single representative experiment.

**Figure 7 viruses-15-00907-f007:**
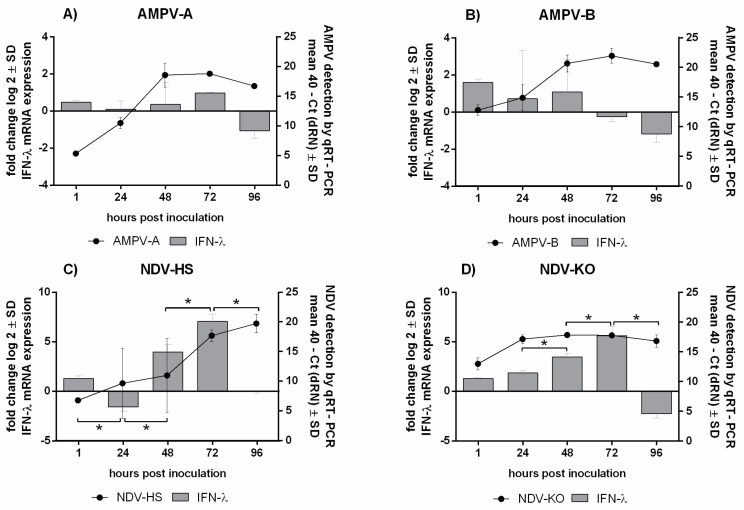
Quantification of IFN-λ mRNA expression in correlation with AMPV and NDV replication in OOCs. (**A**) AMPV-A, (**B**) AMPV-B, (**C**) NDV-HS, and (**D**) NDV-KO. Three rings/group/time points were investigated. Error bars indicate standard deviations. * indicate significant differences between the virus-inoculated group and the virus-free group at the same time point, determined using the Tukey HSD all-pairwise comparison test (ANOVA, *p* < 0.05). The graphs depict data from one representative experiment.

**Figure 8 viruses-15-00907-f008:**
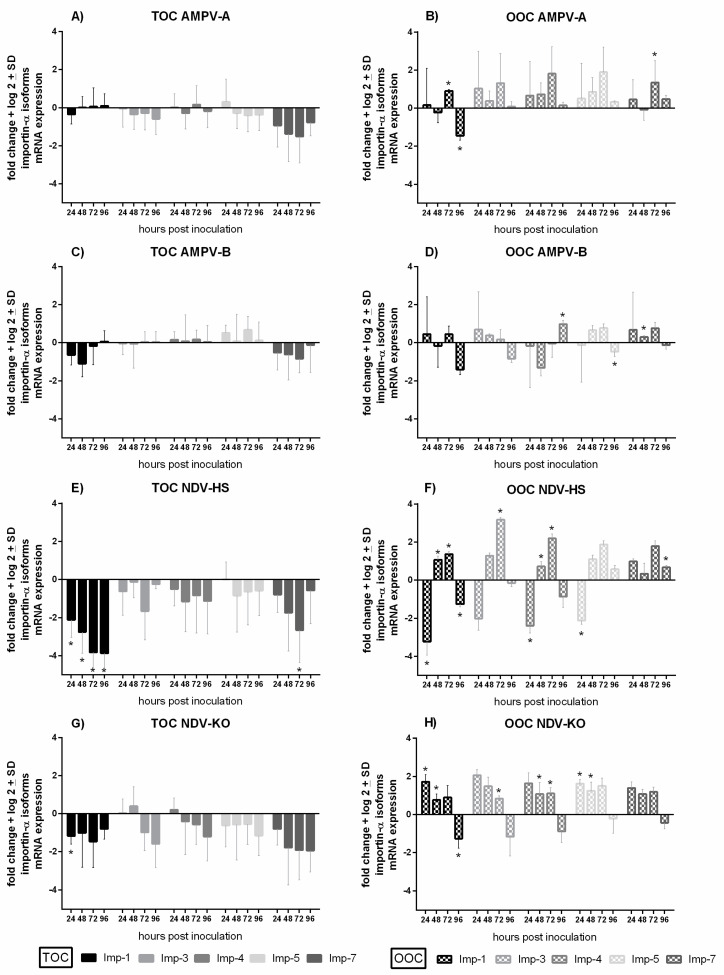
Quantification of mRNA expression levels of importin-α isoforms in TOCs and OOCs after infection with AMPV or NDV. The different mRNA expression patterns of importin-α 1, 3, 4, 5, and 7 were detected after the infection of TOCs (**A**,**C**,**E**,**G**) and OOCs (**B**,**D**,**F**,**H**) with either (**A**,**B**) AMPV-A, (**C**,**D**) AMPV-B, (**E**,**F**) NDV-HS, or (**G**,**H**) NDV-KO. * indicate significant differences between the virus-inoculated group and the virus-free group at the same time point, determined using the Tukey HSD all-pairwise comparison test (ANOVA *p* < 0.05). The graphs represent data from one representative experiment.

## Data Availability

Raw data will be provided upon request.

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
