# Peer review of "In Vitro Investigation of the Interaction of Avian Metapneumovirus and Newcastle Disease Virus with Turkey Respiratory and Reproductive Tissue"

_viruses, 2023, doi:10.3390/v15040907_

Round 1

Reviewer 1 Report

The paper reports results of a complete and exaustive in vitro investigation on tracheal  and  oviduct organ coltures of the behavior and interaction of two main avian viruses affecting turkeys respiratory and reproductive systems. Results obtained demonstrate the suitability of in vivo models to investigate virus host interaction. The paper is clearly written, methods are rigorous and complete, discussion and conclusion supported by results; therefore it can be accepted in the present form.

Author Response

Dear Reviewer,

many thanks for your input. We conducted as suggested a spell check and did some minor adjustments throughout the manuscript.

Reviewer 2 Report

In the current manuscript, the authors investigated the characteristics of AMPV and NDV infection in tracheal and oviduct epithelium using TOC and OOC organoids. Below are the concerns I have with this manuscript:

Concern#1: Introduction sections require re-writing. Authors have simply described their experimental design rather than reviewing the literature. Please add a paragraph describing an etiological agent and associated clinical signs and symptoms for readers unaware of these viral pathogens.

Concern#2: Authors performed two independent experiments but chose to show data originating from one only. Why not show an average of both experiments?

Concern#3: Result sections can be further improved. The citation of figures or sub-figures after sentences is missing, making it difficult for readers to find appropriate figures. Also, include a heading on figures, so that reader can follow it easily instead of looking at the figure legend. Legends are also quite confusing here.

Concern#4: Several times, authors have made a comparison for a virus strain replication in between or within the tissues; however, without those comparison plots. Consider including those comparison plots (in supplementary)

Concern#5: Keep consistency in naming viral strains. In several instances, NDV-Herts 33 is written as NDV-HS in figure legends, but in the test, it is Herts 33.

Concern#6: Line:264, In a case where data are not shown, include (data not shown)

Line 61: Check for the tying mistakes

Line 89-90: Check this sentence; it seems incomplete

Author Response

Dear Reviewer,

many thanks for your useful comments. Below you see an overview on the improvements and comments from our site adressing your concerns.

Concern#1:

Introduction sections require re-writing. Authors have simply described their experimental design rather than reviewing the literature. Please add a paragraph describing an etiological agent and associated clinical signs and symptoms for readers unaware of these viral pathogens.

As suggested, we add two additional paragraphs, one for each virus with additional information about virus taxonomy, strain classification and clinical signs (new version lines 62-69 and 75-84). Therefore we deleted two sentences, old version lines 67-70 (new version 84-87).

Concern#2:

Authors performed two independent experiments but chose to show data originating from one only. Why not show an average of both experiments?

Indeed, we performed two independent experiments. We obtained comparable data from these experiments. To demonstrate this we already had included Figure 2, which also indicates the observed variations between the two experiments. Nevertheless, the pattern and trend of data was always the same between experiments, but slight variations in titer and time course have occurred due to the individual variations, which may be due to variations of the donors. We have to keep in mind that we worked with organ culture systems (for OOC derived from commercial birds, which therefore may have slight variations in their immunological status) and not with a permanent cell line. Therefore, a combination of data of both experiments in one graph is not advisable and leads to  non -interpretable data sets. For theses two reasons, we decided to show only the data of one representative experiment  (with exception for Figure 2) throughout the paper.

Concern#3:

Result sections can be further improved. The citation of figures or sub-figures after sentences is missing, making it difficult for readers to find appropriate figures. Also, include a heading on figures, so that reader can follow it easily instead of looking at the figure legend. Legends are also quite confusing here.

Citation of figures after sentences were added as well as headings in the figures. In case of images representing the results of histology and immunohistochemistry notes are added above or next to the images.

Concern#4:

Several times, authors have made a comparison for a virus strain replication in between or within the tissues; however, without those comparison plots. Consider including those comparison plots (in supplementary)

We add additional graphs for viral replication to compare the different viral strains per organ culture in the supplements (lines 604-606). In case of IFN-lambda we have consciously chosen this kind of presentation to enable a better correlation of IFN-lamda-regulation and viral replication. The importin data presentation was not changes as this selected format provides the best overview. We consider that no additional figure is necessary to support the conclusions.

Concern#5:

Keep consistency in naming viral strains. In several instances, NDV-Herts 33 is written as NDV-HS in figure legends, but in the test, it is Herts 33.

Changes were made in following lines (new version):

118-121 + 130 + 591

Concern#6:

Line:264, In a case where data are not shown, include (data not shown)

`Data not shown´ was added in line 264 (old version); 284 (new version)

Check for the tying mistakes

Line 61 was changed into:

…the infection experiments [11,12], since both may induce lesions in the respiratory as well as in the reproductive tract of turkeys [13-15].

Check this sentence; it seems incomplete

Line 89-90 (old version); line 106-107 (new version) was changed into:

Although NDV and AMPV replicate in the cytoplasm, we assume an interaction for these viruses with importins.

Own minor corrections:

(Lines of new version)

Line 188 Triton TM has been corrected to TritonTM

Line 265 P-value has been corrected to P-value

Line 344 is has been corrected to are

Line 371 space before “As” has been removed

Line 381 one comma has been removed

Line 426 Fig.S1 has been corrected to Fig.S2

Reviewer 3 Report

 This is an interesting manuscript investigating possible differences  between virus infections on  epithelial structures.

The authors compared the interactions of two important poultry viruses on turkey organ cultures. Two members of the order Mononegavirales, Avian Metapneumovirus (AMPV) and Newcastle disease virus (NDV), were selected to conduct the in vitro experiments, since these viruses can infect both, trachea and oviduct. In addition, they used different strains of these viruses, a subtype A- and a subtype B- strain for AMPV and the NDV 24 strains Komarow and Herts`33 to detect possible differences not only between the tissues but also between different viral strains. Turkey tracheal- and oviduct organ cultures (TOC and OOC) were prepared to investigate viral replication, antigen localisation, lesion development and the expression pattern of interferon-λ and importin-α isoforms. All viruses replicated more efficiently in oviduct than in tracheal epithelium (p < 0.05). In addition, the authors observed higher expression levels of both, 29 IFN-λ and importin-α in OOCs compared to TOCs. The results present strain-dependent differences, with AMPV-B- and Herts`33 strain being more virulent in organ cultures than the appropriate AMPV-A- and Komarow- strain, based on the higher viral genome loads, more severe histological lesions and higher upregulation of IFN-λ. 

The introduction is well written describing the importance of this manuscript and explaining the methods used.  Material and methods paragraph ckearly giving the methods used.

The results are clearly documented including figures which showing the experimental data.

The discussion paragraph and the conclusions are clear. 

Author Response

Dear Reviewer,

many thanks for your positive feedback.